# Prevalence and distribution pattern of malaria and soil-transmitted helminth co-endemicity in sub-Saharan Africa, 2000–2018: A geospatial analysis

**Muhammed O. Afolabi** [1] *, **Adekola Adebiyi** [2], **Jorge Cano** [3], **Benn Sartorius** [1,4,5], **Brian Greenwood** [1], **Olatunji Johnson** [6], **Oghenebrume Wariri** [7,8]

1 Department of Disease Control, London School of Hygiene & Tropical Medicine, London, United Kingdom, 2 College of Agriculture, Engineering and Environmental Design, Legacy University, Banjul, The Gambia, 3 Expanded Special Project for Elimination of NTDs, World Health Organization Regional Office for Africa, Brazzaville, Republic of the Congo, 4 Centre for Tropical Medicine and Global Health, University of Oxford, Oxford, United Kingdom, 5 Department of Health Metric Sciences, University of Washington, Seattle, Washington, United States of America, 6 Department of Mathematics, University of Manchester, Manchester, United Kingdom, 7 Department of Infectious Diseases Epidemiology, London School of Hygiene & Tropical Medicine, London, United Kingdom, 8 Vaccines and Immunity Theme, Medical Research Council Unit The Gambia at London School of Hygiene & Tropical Medicine, Fajara, The Gambia

* Muhammed.Afolabi@lshtm.ac.uk

## Abstract

### Background

Limited understanding exists about the interactions between malaria and soil-transmitted helminths (STH), their potential geographical overlap and the factors driving it. This study characterised the geographical and co-clustered distribution patterns of malaria and STH infections among vulnerable populations in sub-Saharan Africa (SSA).

### Methodology/Principal findings

We obtained continuous estimates of malaria prevalence from the Malaria Atlas Project (MAP) and STH prevalence surveys from the WHO-driven Expanded Special Project for the Elimination of NTDs (ESPEN) from Jan 1, 2000, to Dec 31, 2018. Although, MAP provides datasets on the estimated prevalence of *Plasmodium falciparum* at 5km x 5km fine-scale resolution, we calculated the population-weighted prevalence of malaria for each implementation unit to ensure that both malaria and STH datasets were on the same spatial resolution. We incorporated survey data from 5,935 implementation units for STH prevalence and conducted the prevalence point estimates before and after 2003. We used the bivariate local indicator of spatial association (LISA analysis) to explore potential co-clustering of both diseases at the implementation unit levels among children aged 2–10 years for *P. falciparum* and 5–14 years for STH, living in SSA.

Our analysis shows that prior to 2003, a greater number of SSA countries had a high prevalence of co-endemicity with *P.falciparium* and any STH species than during the period from 2003–2018. Similar prevalence and distribution patterns were observed for the co-

**Data Availability Statement:** Open access data collected from Malaria Atlas Project: https://malariaatlas.org/ Expanded Special Project for the

Elimination of Neglected Tropical Diseases: https://espen.afro.who.int/. All other relevant data are within the manuscript and its Supporting Information files.

**Funding:** This study was implemented as part of a career development fellowship awarded to MOA, which is funded under the UK Research and Innovation Future Leaders Fellowship scheme (MR/S03286X/1). The funder had no role in the study design, data collection and analysis, decision to publish, or preparation of the manuscript.

**Competing interests:** The authors have declared that no competing interests exist.

endemicity involving *P.falciparum*-hookworm, *P.falciparum-Ascaris lumbricoides* and *P.falciparum-Trichuris trichiura*, before and after 2003.

We also observed spatial variations in the estimates of the prevalence of *P. falciparum*-STH co-endemicity and identified hotspots across many countries in SSA with inter-and intra-country variations. High *P. falciparum* and high hookworm co-endemicity was more prevalent in West and Central Africa, whereas high *P. falciparum* with high *A. lumbricoides* and high *P. falciparum* with high *T. trichiura* co-endemicity were more predominant in Central Africa, compared to other sub-regions in SSA.

## Conclusions/Significance

Wide spatial heterogeneity exists in the prevalence of malaria and STH co-endemicity within the regions and within countries in SSA. The geographical overlap and spatial co-existence of malaria and STH could be exploited to achieve effective control and elimination agendas through the integration of the vertical control programmes designed for malaria and STH into a more comprehensive and sustainable community-based paradigm.

## Author summary

Malaria and parasitic worms frequently co-exist among children living in the poorest countries of the world, but little is known about the specific locations of the combined infections involving the two major parasitic diseases and how they interact and change over the years.

We used open access data collected by two public registries, that is, the Malaria Atlas Project and Expanded Special Project for the Elimination of NTDs, to understand the overlap of the two diseases in different parts of Africa, where their co-distributions are more predominant.

We found significant differences in the distribution patterns of the combined diseases across different parts of Africa, with large concentrations identified in Central and West Africa. For example, double infections with malaria and hookworm were more common in West and Central Africa, whereas malaria and roundworm, and malaria and whipworm were predominantly found in Central Africa. A large collection of the dual infections was also found in some localities within the countries which appeared to have low burden of the two diseases.

These findings provide a useful insight into the areas which could be serving as a reservoir to propagating the transmission of the two diseases. The results of this study could also be used to develop and implement integrated control programmes for malaria and parasitic worms, and this could help to achieve the WHO NTD roadmap to ending the neglect to attain Sustainable Development Goals by 2030.

## Introduction

Given the environmental and host factors which favour transmission of multiple parasitic infections, malaria and soil-transmitted helminths (STH), including *Ascaris lumbricoides*, *Trichuris trichiura*, and hookworms (*Necator Americanus* and *Ancylostoma duodenale*), co-exist in many parts of the world, predominantly in sub-Saharan Africa (SSA), [1] where they are

most prevalent. An estimated 241 million cases of malaria were reported in 2021 in 85 malaria endemic countries, 29 of which accounted for 96% of malaria cases globally, and six SSA countries (Nigeria, Democratic Republic of the Congo, Uganda, Mozambique, Angola and Burkina Faso) carried more than half of this burden. [2] Similarly, a disproportionately high burden of STH has been reported in many countries in SSA with large concentrations of moderate-to-heavy intensity infections in Nigeria, Democratic Republic of the Congo, Ethiopia, Cameroon, Angola, Mozambique, Madagascar, Equatorial Guinea, and Gabon. [3]

In these malaria-STH co-endemic countries, estimating the national burden of the co-endemicity poses a public health challenge because national surveillance systems are often sub-optimal. [4] For example, deworming through mass drug administration (MDA) of anthelminthic drugs is widely regarded as a cost-effective strategy. However, the impact of these MDA programmes on the changing prevalence and intensity of STH in children, is often hindered by a lack of comprehensive baseline data before MDA initiation, because of financial challenges or as a result of the administration of the drugs via undocumented channels. [3] Nevertheless, recent WHO reports [2,4] indicate that considerable progress has been made in reducing the overall burden of malaria and STH in many countries including some in SSA. To sustain the gains of past decades and move the control efforts towards elimination, WHO has recommended a paradigm shift from disease-specific to an integrated approach. [4]

Empirical studies conducted across co-endemic countries have shown that variations in the prevalence of malaria and STH co-endemicity differed significantly across geographic locations. [5–9] Environmental factors such as temperature and humidity have been identified as the major driving factors for the widespread distributions of parasitic diseases such as malaria and STH. Using a combination model of temperature, rainfall and altitude, Brooker *et al* [1] predicted the continental distribution of each of the major STH species, which suggested that *A. lumbricoides* and *T. trichiura* were prevalent predominantly in equatorial, central and west, and southeast Africa, while the hookworm infection was ubiquitous in every African sub-region. Climate-based distribution models were also used to describe malaria transmission across SSA. [10] These spatial models were subsequently combined to estimate the co-distribution of *Plasmodium falciparum* and hookworm, and identified school-age children as the most vulnerable population for the co-endemicity.

Changes in temperature, humidity, rainfall, and other climatic conditions have an impact on key factors that shape the transmission of malaria, such as the mosquito lifespan and the development of malaria parasites in the vector. [11] Similarly, the interactions between the climatic parameters affect rates of desiccation and death of the eggs of *A. lumbricoides* and *T. trichiura*, and hookworm egg hatching success, thereby affecting the overall survival of the STH parasites and development of infective larval stages. In addition, socio-economic conditions such as agricultural practices and other human behaviours also alter ecological landscape, soil larval loads, vector survival, food security and overall human health and wellbeing, thereby providing favourable interface for transmission and co-distribution of malaria and STH. [12,13] The roles of biological factors driven by immune modulation were also reported to be responsible for the co-existence of the mixed infections. [14]

Given the interplay of several complex factors promoting malaria-STH co-endemicity, limited understanding exists about the spatial distribution of this co-endemicity in vulnerable populations. Majority of the published studies targeting malaria-STH co-endemicity have focused on describing the transmission and burden among affected populations, [5,6,9,15] and have paid little attention to their concurrent spatial distribution. Previous systematic reviews have also reported an over-estimation of the relationship between malaria and helminths making it difficult to establish conclusively the burden of malaria-STH dual infections. [16,17] Consistent with findings of these reviews, a recently conducted systemic review and meta-

analysis of 55 studies which enrolled 37,559 children across low and middle- income countries (LMIC) found a wide variation in the prevalence of malaria-helminth co-endemicity at country level, ranging from 7–76% across LMICs. [18] These findings may be due to the low sensitivity of the diagnostic methods employed for the detection of malaria-helminth co-endemicity in the primary studies included in the systematic review.

The WHO road map for the control and elimination of NTDs for the 2021–2030 period defines STH elimination as a public health problem (EPHP) when there is a prevalence of less than 2% of moderate-to-heavy intensity infections. [4] However, as STH control accelerates and new detection tools are developed, a significant number of people with clinical features outside high intensity infection hotspots are now being seen. [19] The 2020 Population raster of individual countries in SSA from the WorldPop database [20] showed that an estimated 165 million people were in high-risk clusters of *P.falciparum*, hookworm and *A.lumbricoides*. For malaria-STH co-endemicity, about 73 million, 21 million and 118 million people were in high risk clusters of co-endemicity involving *P.falciparum* and hookworm; *P.falciparum* and *T. trichiura*; *P.falciparum* and *A.lumbricoides* respectively. Given that the WorldPop estimates are tied typically to high-risk clusters in a single year, reliable estimates of the transmission dynamics of malaria-STH morbidity hotspots are needed to achieve the NTD roadmap goal of intensifying cross-cutting approaches which integrate STH control with common delivery platforms for diseases that share similar epidemiology such as malaria. Also, empirical evidence shows that obtaining reliable estimates of the distribution of STH-malaria co-endemicity would help address the critical gaps in planning and implementation of integration of STH control strategies with other interventions. [21,22]

We undertook a geospatial analysis of the datasets describing the prevalence of malaria and STH infections among children, obtained from open-access data generated from the Malaria Atlas Project (MAP) [23] and the WHO-driven Expanded Special Project to Eliminate NTDs (ESPEN) [24], to characterise the geographical and co-clustered distribution patterns for malaria and STH co-endemicity among the vulnerable populations in SSA.

## Methods

Based on the classifications of the United Nations Geoscheme for Africa, [25] we categorised SSA into west, east, central and southern African sub-regions. These include 46 of Africa's 54 countries and territories that are fully or partially south of the Sahara, excluding Algeria, Djibouti, Egypt, Libya, Morocco, Somalia, Sudan and Tunisia.

### Data curation

We obtained datasets on the prevalence of *P. falciparum* from the MAP database (last accessed September 2, 2021). MAP is a WHO collaborating platform for geospatial disease modelling which was created for the purpose of determining spatial limits, prevalence and endemicity of *P. falciparum and P. vivax*. MAP provides a database to obtain data on the estimated prevalence of *P. falciparum and P. vivax* at 5km resolution for each year from 2000 onwards.[23] The MAP initiative was established in 2006 to project the expected prevalence and endemicity of malaria in all locations around the world, by modelling the available prevalence data. The MAP described the intensity of malaria transmission or endemicity based on the prevalence of peripheral malaria parasitaemia in children aged 2 to 10 years ($Pf$PR$_{2-10}$). This is because the index has been found to closely correlate with the entomological inoculation rate (EIR) which refers to the number of infectious bites per person per day. Due to the ability of $Pf$PR$_{2-10}$ to mirror the EIR, the MAP generated models to predict the $Pf$PR$_{2-10}$ for a given set of environmental conditions. In high endemicity areas, parasite rate (PR) samples are often restricted to

children aged 2–10 years, but in areas of low endemicity, surveys are usually extended to include all age groups. [26,27]

MAP collated survey data that are typically clustered at village level and recorded data on parasite positivity rates. MAP used data points generated from malaria surveys for its estimation by combining these data points with Geographic Information System (GIS) data. MAP was updated in 2010 to include new malariometric data extending to 13,449 administrative units in 43 endemic countries and 22,212 *P. falciparum* parasite rate surveys were used to define spatial limits of malaria transmission. To generate continuous global maps, MAP matched spatially referenced data, e.g., altitude, temperature, rainfall, and vegetation extent with the PR-survey positions, to establish uni-and/or multivariate statistical relationships between malaria endemicity and the environmental factors that affect the distribution of the mosquito vectors responsible for malaria transmission. [23] We focused on *P. falciparum* in this analysis because it is the predominant causative agent for more than 95% cases of clinical malaria in SSA. [2]

We also obtained the datasets on the prevalence of STH (*A. lumbricoides*, *T. trichiuria* and hookworms) from the WHO-driven ESPEN portal. ESPEN portal contains survey datasets on NTDs in Africa.[24] Sartorius and colleagues [3] developed a Bayesian spatio-temporal hierarchical model to analyse the STH datasets from 2000 to 2018; and provided the prevalence estimates at implementation unit (IU) covering the entire period. We used these estimates to determine the hotspots of *P.falciparum*-STH co-endemicity. Given that many control programmes implemented during the period under review could have changed the infection risk for each of the STH species, we took the temporal effects of control programmes into consideration to reflect the transmission hotspots before and after the scale-up of the control programmes. As many control programmes were scaled up around 2003, we conducted the point prevalence estimates before 2003 and from 2003–2018.

To ensure that both the malaria and the STH datasets were on the same spatial resolution, we calculated the population-weighted prevalence of malaria for each IU using the population density data obtained from the WorldPop database. [20] WHO consider the IU as the geographical area over which a particular treatment strategy or intervention is applied. This geographical area could be a district, local government area, county, or province. In most SSA countries, the IU is the second administrative level and there were a total of 5,935 IUs in the study areas reported in this paper. [23]

## Statistical analysis

**Spatial autocorrelation.** To examine the spatial autocorrelation between malaria and STH, we used the bivariate Moran's I statistics (https://pro.arcgis.com/en/pro-app/latest/tool-reference/spatial-statistics/spatial-autocorrelation.htm). We showed via a scatter plot the degree to which the prevalence of a given variable at a given IU is correlated with its neighbours for a different variable. Hence, we plotted the prevalence of *P. falciparum* on the x-axis against the spatial lag of other variables including hookworm, *T. trichiura* and *A. lumbricoides* on the y-axis. The variables were standardised before generating the Moran's I scatter plot using the z-scores. The z-scores enabled us to compare the prevalence estimates of any two variables as they might have different mean and a standard deviation. Z-score value equal to zero was interpreted as the mean prevalence; above zero as higher than the mean prevalence and below zero as lower than the mean prevalence. Therefore, on the scatter plots, all points with values above the mean (i.e. above zero) were IUs with higher than mean prevalence while all points with values below the mean (i.e. below zero) were IUs with lower than mean prevalence. The scatter plot was divided into four quadrants; upper-right, lower-left, upper-left and

lower-right. The values of *P. falciparum* and the spatial lag of any of hookworm, *T. trichiura* and *A.lumbricoides* that fell in the upper-right quadrant and the lower-left quadrant were regarded as an evidence of positive spatial autocorrelation while those in other quadrants indicate negative spatial autocorrelation. The overall conclusion of positive, negative or no spatial autocorrelation depends on the dominant quadrant and the overall Moran's I estimates. A positive value of Moran's I estimate indicates positive spatial autocorrelation while a negative value indicates negative spatial autocorrelation (https://arcview-gis.software.informer.com/10.3/).

**Bivariate spatial clustering.** To identify the spatial clustering of the IUs, we used local Moran's I algorithm in Geoda software (https://geodacenter.github.io/) to implement the bivariate local indicator of spatial association (LISA). [28] Bivariate LISA is a method that uses local spatial autocorrelation to identify localised IUs where prevalence values of one variable are strongly positively or negatively associated with spatial lag of another variable. It gives an indication of the extent of significant spatial clustering of similar values around that observation. We used LISA to identify the hotspots and coldspots, and areas endemic to malaria-STH co-endemicity across the study areas. We considered alpha level ≤0.001 as statistically significant hotspots.

## Results

Overall, a high prevalence of *P.falciparum* and hookworm co-endemicity was reported in more countries, compared to co-endemicity involving *P. falciparum* and *A. lumbricoides*; or *P. falciparum* and *T. trichiura*, which were found in lesser number of countries. Similarly, a high prevalence of co-endemicity and significant hotspot locations were generally predominant in more countries pre-2003 compared to 2003–2018 (S1–S3 Tables).

### *P. falciparum* and co-endemicity with any species of soil-transmitted helminths

A high prevalence of co-endemicity of *P. falciparum* and any STH species was observed in 20 sub-Saharan African countries pre-2003, in contrast to only 12 countries from 2003–2018. These were mostly in West and Central Africa, pre and post 2003. The countries with locations with a high prevalence of co-endemicity of *P. falciparum* and any STH species pre-2003 were Benin, Cote d'Ivoire, Sierra Leone, Liberia, Guinea, Togo, Nigeria, Cameroon, Central Africa Republic, Equatorial Guinea, Gabon, Congo, Angola, Tanzania, Democratic Republic of the Congo (DRC), Burundi, Uganda, Zambia, Mozambique, Madagascar. Countries with locations with a high prevalence of co-endemicity of *P. falciparum* and any STH species from 2003–2018 were Benin, Cote d'Ivoire, Guinea, Liberia, Nigeria, DRC, Cameroon, Equatorial Guinea, Gabon, Congo, Angola, Mozambique (Fig 1 and S1 Table).

There were significant (p<0.001) hotspot locations of *P. falciparum* and any STH co-endemicity in six countries pre-2003, and 2003–2018 respectively, with positive spatial autocorrelation showing Moran's I = 0.241 (pre-2003), and 0.256 (2003–2018), respectively (S1 Fig). The six countries with significant hotspot locations pre-2003 were Cameroon, Equatorial Guinea, Gabon, and Angola in central African, Tanzania in east Africa, and Madagascar in southern Africa. The six countries with significant hotspot locations from 2003–2018 were Liberia and Nigeria in west Africa and Cameroon, Equatorial Guinea, Gabon, and Congo in central Africa (Fig 1 and S1 Table).

### *P. falciparum* and hookworm co-endemicity

There was a high prevalence of co-endemicity of *P. falciparum* and hookworm in 19 sub-Saharan African countries pre-2003, in contrast to 16 countries from 2003–2018. Pre-2003, 52% of

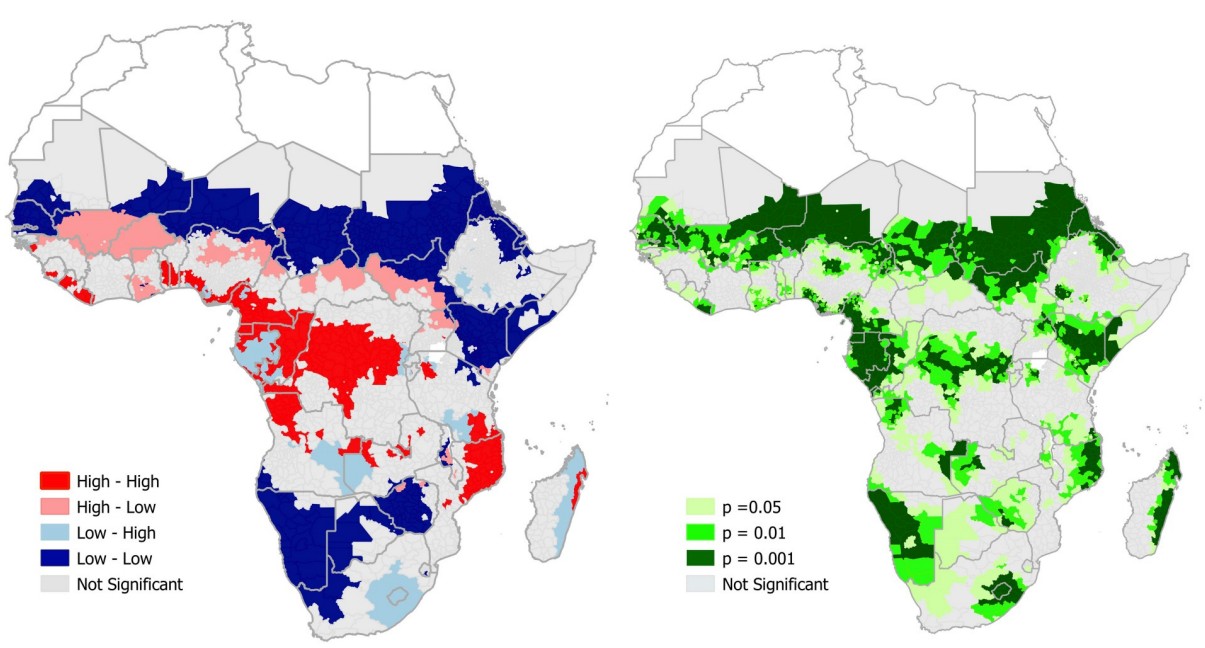

## Pre 2003

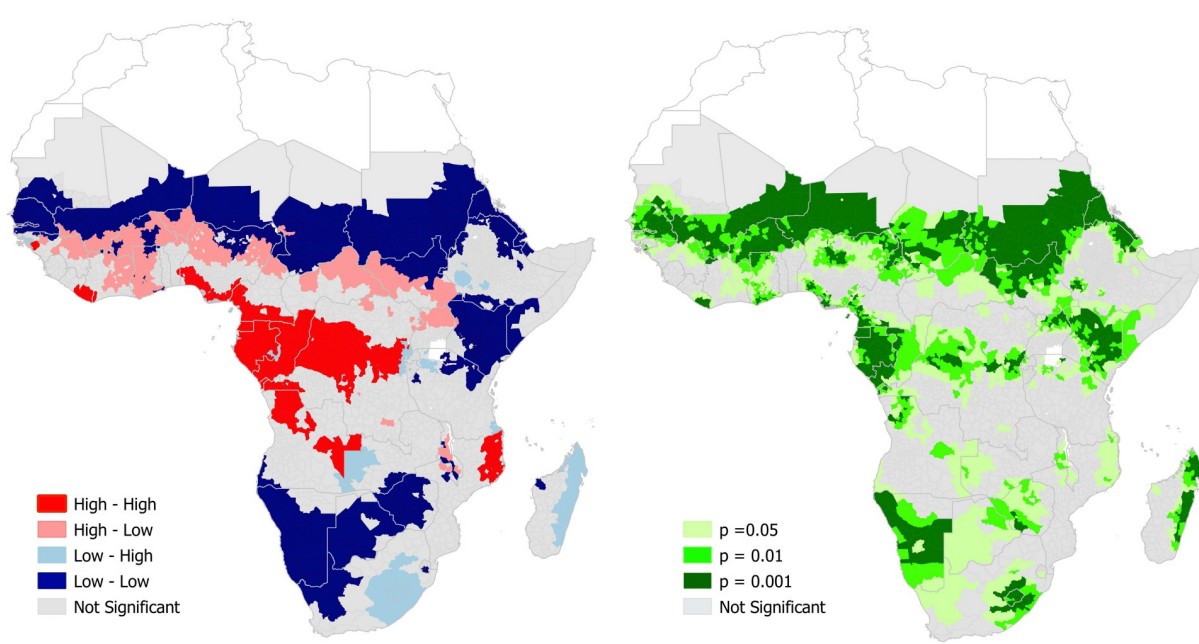

## 2003 - 2018

**Fig 1. The prevalence of *P. falciparum* and any soil-transmitted helminth co-endemicity, and significant hotspots of *P. falciparum* and any soil-transmitted co-endemicity in sub-Saharan Africa.** The density description next to the colour-coded box refers to the prevalence of *P. falciparum*, whilst the column further to the right refers to density of any STH. Link to the base layer of the map: https://gadm.org/index. html.

the 19 countries with a high prevalence of co-endemicity of *P. falciparum* and hookworm were in west Africa, while from 2003–2018, there were 43% (7/16) in west Africa and 31.3% (5/16) in central Africa. The 19 countries with locations with a high prevalence of co-endemicity of *P. falciparum* and hookworm pre-2003 were Benin, Cote d'Ivoire, Ghana, Guinea, Guinea Bissau, Liberia, Nigeria, Togo, Sierra Leone, Cameroon, DRC, Equatorial Guinea, Gabon, Uganda, Tanzania, Angola, Mozambique, Madagascar, and Zambia (Fig 2 and S1 Table). Of these 19 countries pre-2003, there were significant (p<0.001) hotspot locations of *P. falciparum* and hookworm co-endemicity in seven countries, with positive spatial autocorrelation, and Moran's I = 0.390 (S2 Fig). The seven countries with significant hotspot locations were Liberia, Nigeria, Ghana, Sierra Leone, Angola, Uganda and Zambia.

The 16 countries with locations of high prevalence of co-endemicity of *P. falciparum* and hookworm in the period 2003–2018 were Benin, Cote d'Ivoire, Nigeria, Togo, Ghana, Liberia, Guinea, Angola, Cameroon, DRC, Equatorial Guinea, Gabon, Uganda, Tanzania, Zambia and Mozambique. Among these 16 countries, there were significant (p<0.001) hotspot locations of *P. falciparum* and hookworm co-endemicity in Guinea and Nigeria, both in west Africa (Fig 2 and S1 Table), with positive spatial autocorrelation and Moran's I = 0.245 (S2 Fig).

## *P. falciparum* and *A. lumbricoides* co-endemicity

There was a high prevalence of co-endemicity of *P.falciparum* and *A.lumbricoides* in 12 sub-Saharan African countries pre-2003, in contrast to 11 countries from 2003–2018. Pre-2003, 58% of the 12 countries with high prevalence of co-endemicity of *P. falciparum* and *A.lumbricoides* were in central Africa, while from 2003–2018, 63.6% (7/11) of the countries were in central Africa. The 12 countries with locations of high prevalence of *P. falciparum* and *A. lumbricoides* co-endemicity pre-2003 were Liberia, Nigeria, Cameroon, Equatorial Guinea, Gabon, Congo, Angola, Burundi, DRC, Kenya, Mozambique and Madagascar. The 11 countries with locations with a high prevalence of co-endemicity of *P. falciparum* and *A.lumbricoides* in the period 2003–2018 were Liberia, Nigeria, Cameroon, DRC, Equatorial Guinea, Gabon, Congo, Angola, Burundi, Kenya, Mozambique (Fig 3 and S1 Table).

Pre-2003, there were significant (p<0.001, positive spatial autocorrelation, with Moran's I = 0.059) hotspot locations of *P. falciparum* and *A.lumbricoides* co-endemicity in six countries, compared to only three countries during the period 2003–2018 (p<0.001, Moran's I = 0.184) (S3 Fig). The six countries pre-2003 were Liberia, Cameroon, Equatorial Guinea, Angola, Burundi and Madagascar. The three countries with significant hotspots from 2003–2018 were Cameroon, Equatorial Guinea, and Angola, all located in central African region (Fig 3 and S1 Table).

## *P. falciparum* and *T. trichiura* co-endemicity

Pre-2003, there was a high prevalence of co-endemicity of *P. falciparum* and *T. trichiura* in 10 sub-Saharan African countries, mainly located in central Africa (70%; 7/10). The 10 central African countries with locations of high co-endemicity of *P. falciparum* and *T.trichiura* pre-2003 were Liberia, Cameroon, Equatorial Guinea, Gabon, Congo, DRC, Angola, Burundi, Kenya, and Madagascar. Among these 10 countries, there were significant hotspot locations (p<0.001, with positive spatial autocorrelation, Moran's I = 0.037) of *P. falciparum* and *T. trichiura*, (S4 Fig) in Cameroon, Equatorial Guinea, Gabon, and Madagascar, (Fig 4 and S1 Table).

From 2003–2018, there was a high prevalence of co-endemicity of *P. falciparum* and *T. trichiura* in only five sub-Saharan African countries (mainly in central Africa). The five central African countries with locations of high co-endemicity of *P. falciparum* and *T. trichiura* from

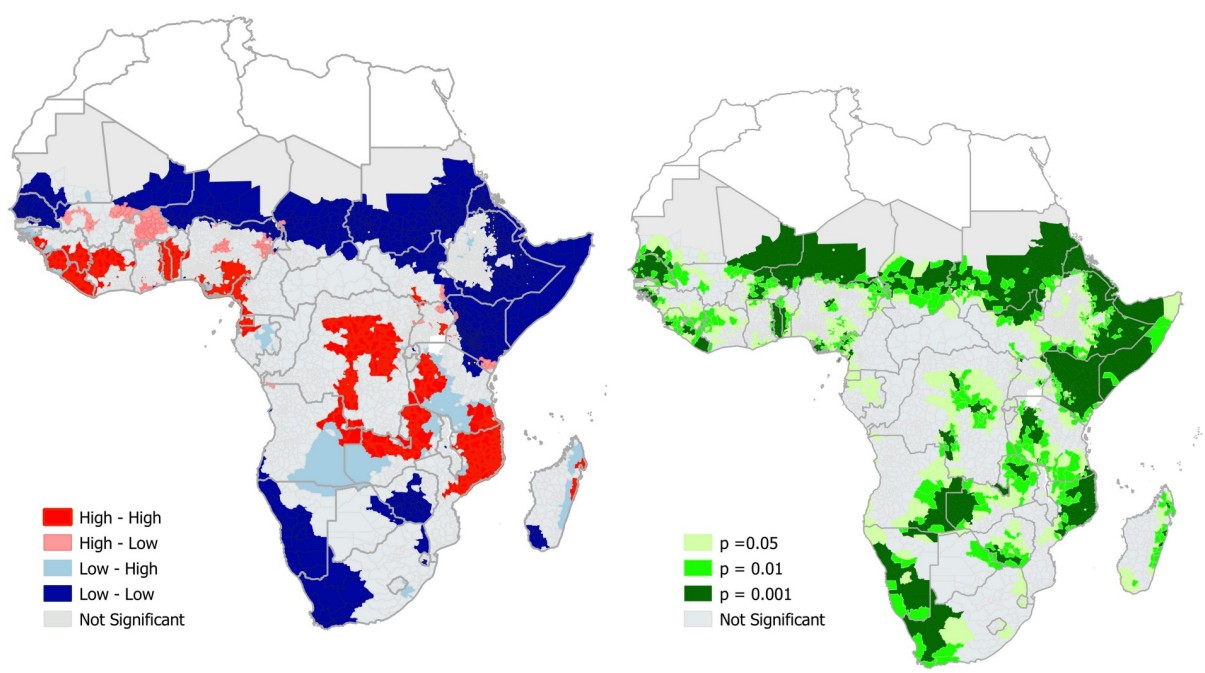

Pre 2003

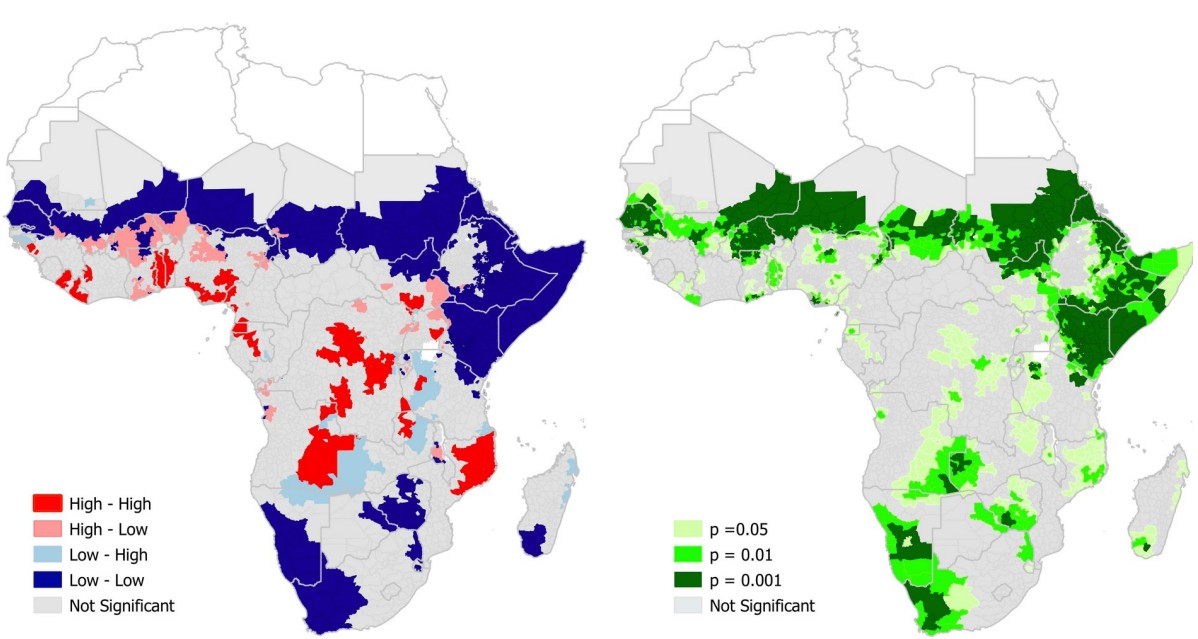

2003 - 2018

**Fig 2. The prevalence of *P. falciparum* and hookworm co-endemicity, and significant hotspots of *P.falciparum* and hookworm co-endemicity in sub-Saharan Africa.** ʼThe density description next to the colour coded box refers to the prevalence of *P. falciparum*, whilst the column further to the right refers to density of hookworm. Link to the base layer of the map: https://gadm.org/index.html.

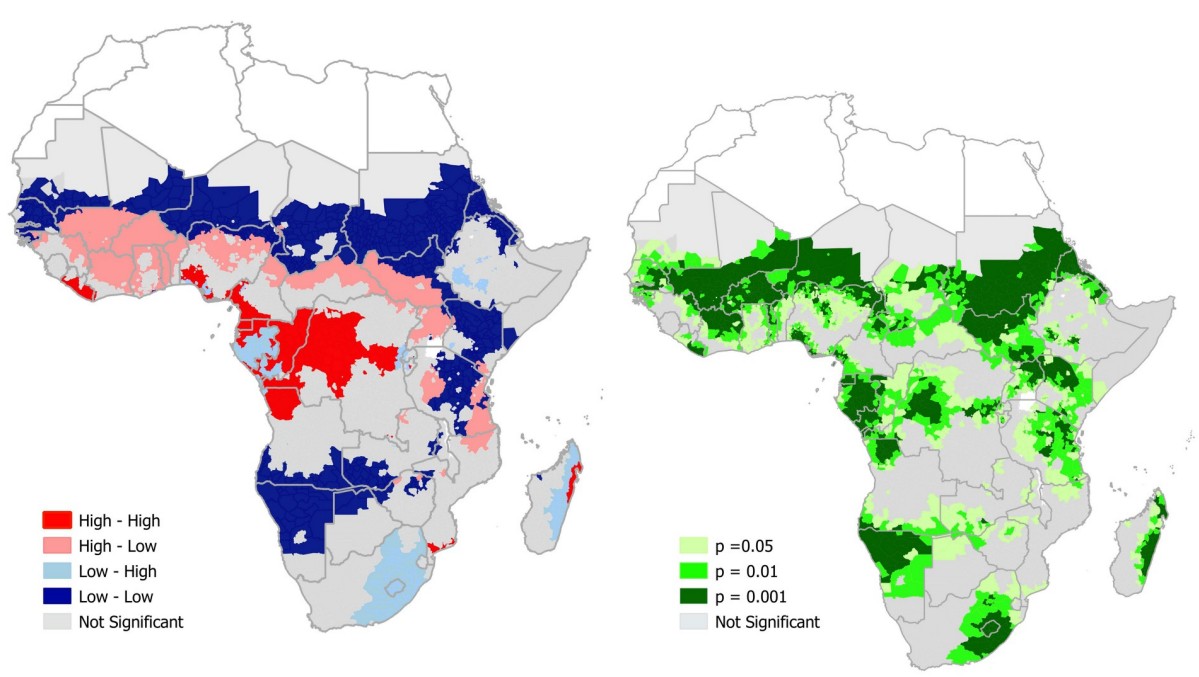

Pre 2003

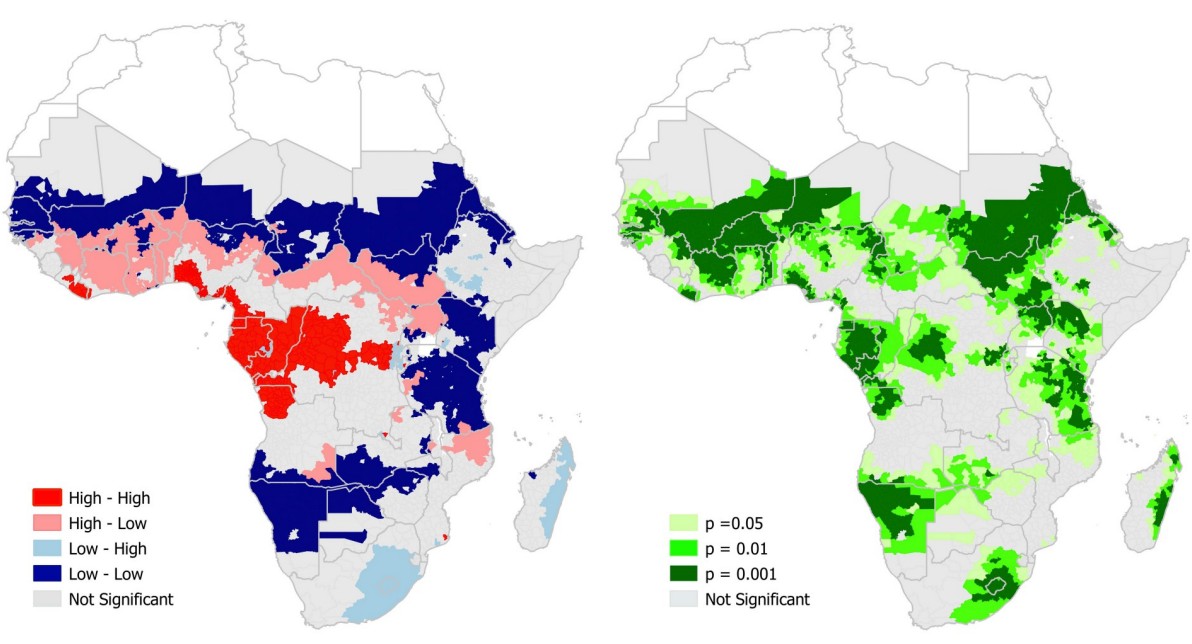

2003 - 2018

**Fig 3. The prevalence of *P. falciparum* and *Ascaris lumbricoides*, and significant hotspots of *P. falciparum* and *A.lumbricoides* co-endemicity in sub-Saharan Africa.** The density description next to the colour-coded box refers, to the prevalence of *P. falciparum* whilst the column further to the right refers to density of *A. lumbricoides*. Link to the base layer of the map: https://gadm.org/index.html.

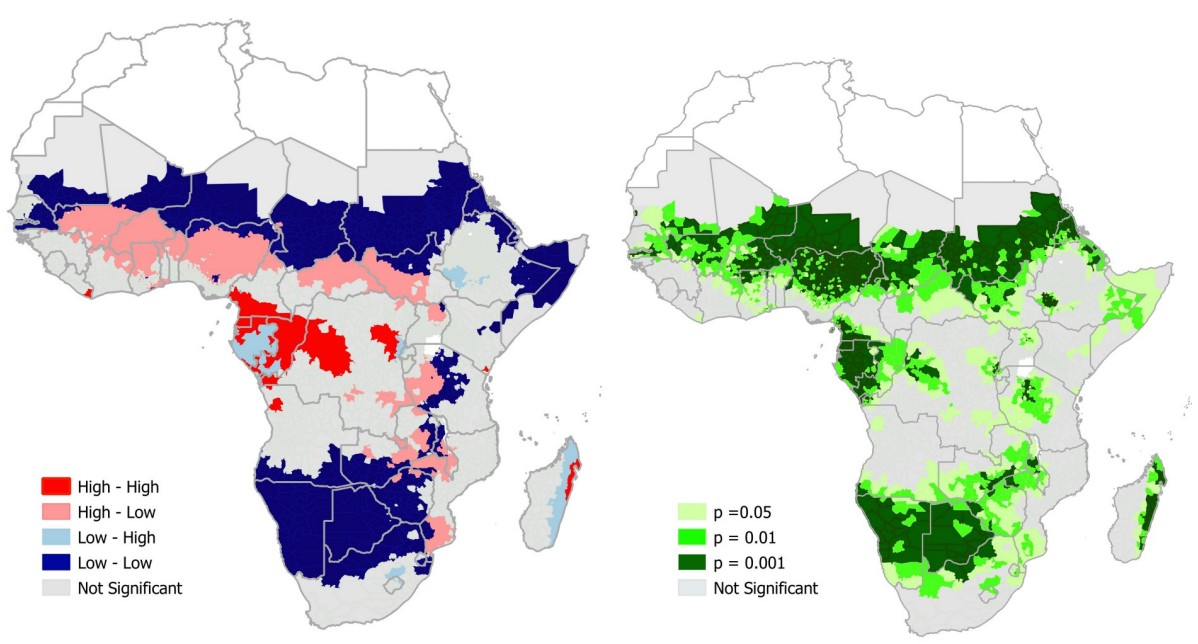

Pre 2003

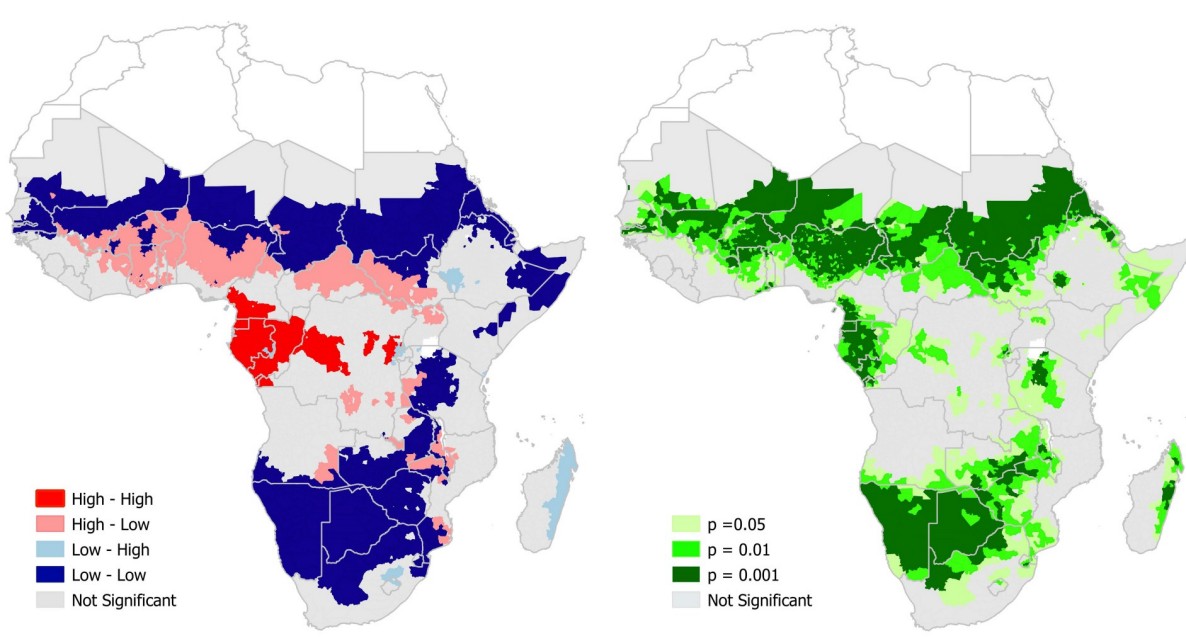

2003 - 2018

**Fig 4. The prevalence of *P. falciparum* and *T.trichiura*, and significant hotspots of *P.falciparum* and *T. trichiura* co-endemicity in sub-Saharan Africa.** The density description next to the colour-coded box refers, to the prevalence of *P. falciparum* whilst the column further to the right refers to density of *T.trichiura*. Link to the base layer of the map: https://gadm.org/index.html.

2003–2018 were Cameroon, Equatorial Guinea, Gabon, Congo, DRC. Among these five countries, there were significant hotspot locations (p<0.001, with positive spatial autocorrelation, Moran's I = 0.069) of *P. falciparum* and *T. trichiura* (S4 Fig) in Cameroon, Equatorial Guinea, Gabon (Fig 4 and S1 Table).

## Discussion

Our analysis shows spatial variations in the estimates of prevalence of *P.falciparum*-STH co-endemicity and identified hotspots across many countries in SSA where the co-endemicity were high. Before the scale-up of specific control programmes for malaria and STH in 2003, our findings showed that a greater number of countries in SSA had a high prevalence of co-endemicity with *P.falciparium* and any STH species than during the period from 2003–2018 when the control programmes were expanded. Similar prevalence and distribution patterns were observed for the co-endemicity involving *P.falciparum*-hookworm, *P.falciparum-A.lumbricoides* and *P.falciparum-T.trichiura*, during these periods.

Consistent with previous studies [3,17,29,30], our findings also highlight inter- and intra-country variations in the prevalence of malaria-helminth co-endemicity. Across Central and West Africa sub-regions, a high prevalence of *P.falciparum* and high hookworm co-endemicity was predominant, while the burden of *P. falciparum-A. lumbricoides* and *P.falciparum-T. trichiura* co-endemicity was higher in Central Africa than in neighbouring sub-regions. Although, recent reports have shown that a considerable gain has been made in the control of malaria and STH in the last two decades, [1,19] our findings suggest that a high burden of falciparum malaria and moderate-to-heavy infections for at least one STH species are still occurring in the southern part of West Africa and extending to Central Africa including Cameroon, Burundi and DRC. We observed similar patterns and trends in the hotspot distributions for *P. falciparum* and any of the STH species. While programmatic implementation of preventive chemotherapy and intermittent preventive treatment of malaria in children (IPTc) which later became seasonal malaria chemoprevention (SMC) have contributed to a significant reduction in the burden of STH and malaria in many countries, inherent challenges such as logistical burden, medication adherence, and incomplete coverage have been reported to be responsible for the persistently high burden of malaria and STH in the spatial areas identified in our study. [19,21]

Taken together the geographical overlap and spatial co-existence of malaria and STH, favoured largely by environmental factors [1,3], could be exploited to achieve effective control and elimination agendas through integration of the vertical programmes designed for malaria and STH into a more comprehensive and sustainable community-based paradigm, that includes other preventive care such as hand hygiene and water sanitation. The WHO 2030 NTD road map recommends concrete actions within integrated platforms for delivery of the interventions needed to improve the cost–effectiveness, coverage and geographical reach of these integrated programmes. [4] A successful integration of the vertical control programmes aligns well with the global trend towards integrated, non-disease specific approaches which have become an increasingly recognised strategy recommended by WHO. [22] However, limited evidence is currently available on the feasibility and effectiveness of integrating malaria with STH control at community level and evidence that this can be done effectively is needed.

Our study has some limitations. First, our study used secondary data obtained from two different sources. The datasets on *P.falciparum* obtained from Malaria Atlas Project were generated from statistical approaches to modelling the prevalence of malaria on a global scale using Bayesian model-based geostatistics, while STH datasets obtained from ESPEN were direct estimates of the prevalence of STH. We overcame the uncertainty posed by this challenge by

ensuring that both malaria and STH datasets were on the same spatial resolution by calculating the population-weighted prevalence of malaria for each implementation unit using the population-density data obtained from the WorldPop database. Furthermore, it is a common occurrence in spatial epidemiology that very limited number of measurements were done in large parts of the geographical region of interest, thereby reporting error-prone or incomplete measurements. Nevertheless, the use of Bayesian model-based geostatistics addressed the spatial variations in the level of uncertainty associated with the mapped surfaces.

Another limitation of our study is that the STH datasets focused largely on school-age children while the malaria datasets concentrated predominantly on children aged 2–10 years because of the ability of prevalence of peripheral malaria parasitaemia in children aged 2–10 years ($Pf$PR$_{2-10}$) to mirror the entomological inoculation rate, making the MAP to generate models to predict the $Pf$PR$_{2-10}$ for a given set of environmental conditions. [27] While the prevalence of *P.falciparum*-STH co-endemicity is known to vary in a relatively predictable fashion in space and time, the observed prevalence depends heavily on individual's age and intervention coverage. The disparity in the ages of children and relatively higher focus on school age children may explain the inconsistencies of our findings with empirical studies, especially for STH prevalence. [31,32] For example, we observed high uncertainty around estimates in Central Africa and some parts of West Africa, which corresponds to areas with a high burden of infection. Lack of survey data for corresponding age groups is a plausible reason for this observation. [3]

More importantly, the absence of survey data or inadequacy in definition of space-time that contributes to the uncertainty observed in our findings may have also introduced bias leading to over-estimation or under-estimation of the prevalence of STH and malaria co-endemicity. Contributing further to the likely over or under-estimation of the burden of STH and malaria could be empirical treatments of malaria and STH outside the control programmes. Our findings could also be affected by the diagnostic approaches involving malaria rapid testing [33] and Kato-Katz method [34] used to generate the data, as these have been widely reported to be less sensitive than molecular methods such as PCR, especially in low-transmission settings.

In conclusion, we have demonstrated wide spatial heterogeneity within the SSA and within countries, on the prevalence of malaria and STH confections. To consolidate on the encouraging progress made in the reduction of the prevalence of malaria and STH in some SSA countries, it is crucial that integrated control programmes target the regions with high prevalence of malaria-STH co-endemicity and areas of hotspot transmission. Our study may also have critical implications for policy-making and resource allocations based on the needs of each region, as well as for the future of the implementation of integrated malaria-helminth control programmes. Nevertheless, it is important that our findings are confirmed by further empirical studies as this will provide the platform for research agenda that will lead to the establishment of the much-needed platform for the implementation of the integrated programmes that are cost-effective, and make optimum use of limited human resources frequently found in SSA.

## Supporting information

**S1 Fig. Moran's I statistic plot showing spatial auto-correlation for *P. falciparum* vs any soil-transmitted helminth co-endemicity in sub-Saharan Africa, pre-2003 and 2003–2018.** (TIF)

**S2 Fig. Moran's I statistic plot showing spatial auto-correlation for *P. falciparum* vs hookworm co-endemicity in sub-Saharan Africa, pre-2003 and 2003–2018.** (TIF)

**S3 Fig. Moran's I statistic plot showing spatial auto-correlation for *P. falciparum* vs *Ascaris lumbricoides* co-endemicity in sub-Saharan Africa, pre-2003 and 2003–2018.**
(TIF)

**S4 Fig. Moran's I statistic plot showing spatial auto-correlation for *P. falciparum* vs *T.trichiura* co-endemicity in sub-Saharan Africa, pre-2003 and 2003–2018.**
(TIF)

**S1 Table. The prevalence of P. falciparum and soil-transmitted helminth (Hookworm, *Ascaris lumbricoides, Trichiuris trichiura*) co-endemicity, and their corresponding significant hotspots locations (ADM1, ADM2, and ADM3) in sub-Saharan Africa, pre-2003 and 2003–2018.**
(DOCX)

**S2 Table. Summary table of STH prevalence estimates by country and frequency of prevalence surveys per country, sub-Saharan Africa, 2000–2018.**
(DOCX)

**S3 Table. Summary table of *P.falciparum* prevalence estimates by country and frequency of prevalence surveys per country, sub-Saharan Africa, 2000–2018.**
(DOCX)

## Acknowledgments

We thank Mike Thorn of the Malaria Atlas Project and Ismaela Abubakar for supporting the data curation for this study.

## Author Contributions

**Conceptualization:** Muhammed O. Afolabi.

**Data curation:** Muhammed O. Afolabi, Jorge Cano, Benn Sartorius, Olatunji Johnson.

**Formal analysis:** Muhammed O. Afolabi, Adekola Adebiyi, Olatunji Johnson.

**Funding acquisition:** Muhammed O. Afolabi.

**Investigation:** Muhammed O. Afolabi, Oghenebrume Wariri.

**Methodology:** Muhammed O. Afolabi.

**Project administration:** Muhammed O. Afolabi, Brian Greenwood, Oghenebrume Wariri.

**Resources:** Muhammed O. Afolabi, Brian Greenwood, Oghenebrume Wariri.

**Supervision:** Muhammed O. Afolabi, Brian Greenwood.

**Validation:** Adekola Adebiyi, Jorge Cano, Benn Sartorius, Brian Greenwood, Olatunji Johnson.

**Visualization:** Muhammed O. Afolabi, Adekola Adebiyi, Jorge Cano, Benn Sartorius, Brian Greenwood, Olatunji Johnson, Oghenebrume Wariri.

**Writing – original draft:** Muhammed O. Afolabi, Olatunji Johnson, Oghenebrume Wariri.

**Writing – review & editing:** Muhammed O. Afolabi, Adekola Adebiyi, Jorge Cano, Benn Sartorius, Brian Greenwood, Olatunji Johnson, Oghenebrume Wariri.

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
