## [Decision Letter · Decision Letter 0]

22 Apr 2022

Dear Dr. Afolabi,

Thank you very much for submitting your manuscript "Estimating the burden of malaria and soil-transmitted helminth co-infection in sub-Saharan Africa: a geospatial study" for consideration at PLOS Neglected Tropical Diseases. As with all papers reviewed by the journal, your manuscript was reviewed by members of the editorial board and by several independent reviewers. In light of the reviews (below this email), we would like to invite the resubmission of a significantly-revised version that takes into account the reviewers' comments. 

We cannot make any decision about publication until we have seen the revised manuscript and your response to the reviewers' comments. Your revised manuscript is also likely to be sent to reviewers for further evaluation.

Sincerely,

Uwem Friday Ekpo, PhD

Associate Editor

Marco Coral-Almeida

Deputy Editor

Reviewer's Responses to Questions

**Key Review Criteria Required for Acceptance?**

**Methods**

-Are the objectives of the study clearly articulated with a clear testable hypothesis stated?

-Is the study design appropriate to address the stated objectives?

-Is the population clearly described and appropriate for the hypothesis being tested?

-Is the sample size sufficient to ensure adequate power to address the hypothesis being tested?

-Were correct statistical analysis used to support conclusions?

-Are there concerns about ethical or regulatory requirements being met?

Reviewer #1: METHODS

- Is the MAP data only individuals aged 5-14 years? Clarify. More information is needed on the prevalence estimates for MAP; provide key information on demographics, source of prevalence surveys, and concentration of data by country.

- Provide summary tables (can go in SI) for which years STH prevalence estimates were available by country and how many prevalence surveys per country were available, as the sparsity of the data will influence your estimates. Ideally, something similar for the MAP estimates should be shown too.

- You are using estimates from MAP yet direct prevalence measurements from ESPEN. Why? Please explain how the methods for MAP and methods for STH are the same or differ in the prevalence kriging or estimation. If MAP data is not from prevalence surveys then discuss what the key dominating environmental features are of that data and how those environmental features feature into STH estimation (if at all). The introduction would have already given us an idea of the importance of each feature for each set of infections.

Reviewer #2: -Are the objectives of the study clearly articulated with a clear testable hypothesis stated? No applicable

-Is the study design appropriate to address the stated objectives? Partially

-Is the population clearly described and appropriate for the hypothesis being tested? No applicable

-Is the sample size sufficient to ensure adequate power to address the hypothesis being tested? No applicable

-Were correct statistical analysis used to support conclusions? Partially

-Are there concerns about ethical or regulatory requirements being met? No

The authors estimated the overall prevalence for each species over 18 years (i.e. 2000-2018). However, due to many control programs done in this period, the infection risk for each species changed a lot over the study period, temporal effect should not be ignored. As MAP provides estimate prevalence of P. falciparum for each year, and ESPEN provides STH survey data from 2000 to 2018, the data which the analysis based includes temporal information. Advanced methods should be used to take into account temporal effect.

The authors used the inverse distance weighted kriging methods to produce STH prevalence surface. However, they did not provide any further methodology on how to validate the model performance. For example, how to assess the accuracy of the method on estimating the prevalence where no survey data exist? In addition, as the prevalence surfaces were produced from spatial interpolation instead of real observation, methods and results for prediction uncertainty should be provided.

In line 164, please explain what “spatially lagged variables” means.

In lines 165-167, the authors define high prevalence and low prevalence as values above and below the mean, respectively. What is the rationality for this definition? For example, if the prevalence for one species is very high across the study region, e.g., the prevalence of all areas is above 50%, then defining the areas with prevalence lower the mean level as “low prevalence” areas seems not reasonable. Besides, was the prevalence standardized before analysis? All prevalence should be above or equal to zero, however, the authors used the “above zero” and “below zero” in the definition.

In lines 188-189, how to define the “high-risk clusters of P. falciparum, hookworm and A. lumbricoides”? Besides, the numbers of people in these clusters should be put in the result section instead in the methodology section.

**Results**

-Does the analysis presented match the analysis plan?

-Are the results clearly and completely presented?

-Are the figures (Tables, Images) of sufficient quality for clarity?

Reviewer #1: (No Response)

Reviewer #2: -Does the analysis presented match the analysis plan? Partially

-Are the results clearly and completely presented? No

-Are the figures (Tables, Images) of sufficient quality for clarity? No

The authors provide results for co-infection of P. falciparum and each species of STH. Results for co-infection of malaria and any species of STH seems more meaningful for integrated control programs, as control methods for the three species of STH are similar. Such results should also be provided.

The authors put the estimated prevalence of each species in the attachment. As these results are important, they should be put in the main text and more description of these results should be provided.

In Figs 1A-1C, the prevalence below “-1” seems cut. What is the reason for this? Besides, how could prevalence below zero?

**Conclusions**

-Are the conclusions supported by the data presented?

-Are the limitations of analysis clearly described?

-Do the authors discuss how these data can be helpful to advance our understanding of the topic under study?

-Is public health relevance addressed?

Reviewer #1: DISCUSSION

- A major limitation of the study is the 5km resolution, especially for STHs.

- Information is needed on the effect of key ecological features in the estimations. Discuss gradients of infection prevalence for both malaria and hookworm as one moves away from water bodies; this will affect the interpolation. 

- Rerun the analysis from 2003- onwards as this period is when MDA was substantially scaled up and discuss whether there is any heterogeneity pre 2003 versus post. Also, the breakdown of data pre-post MDA and its discussion is needed.

- Discuss limitations of focusing on school-aged children, as hookworm at least is known to increase in prevalence/intensity with age.

Reviewer #2: -Are the conclusions supported by the data presented? Partially

-Are the limitations of analysis clearly described? Partially

-Do the authors discuss how these data can be helpful to advance our understanding of the topic under study? Yes

-Is public health relevance addressed? Yes

**Editorial and Data Presentation Modifications?**

Reviewer #1: (No Response)

Reviewer #2: Attentions should be paid to the writing, as following:

In many places of the manuscript, species was not written italic, and there is no space between the first and the second words of the species. For example, errors in line 231.

Presentation of the figures should also be revised. For example, Fig 1 should be a figure instead of separating it into Fig 1A, Fig 1B and Fig 1C. Please follow the standards of the journal.

The positions of the titles of figures should also be revised according to the journal standards. For example, the positions of Fig 3A should be below the figure.

Authors should be carefully check the typing errors. For example, in line 209, does “high prevalence of P. falciparum and high HW co-infection” mean “high prevalence of P. falciparum and HW co-infection”? Besides, if “HW” is the abbreviation of hookworm, it should be mentioned at the first time it appears.

In fig 2B, 3B and 4B, “probability” should be revised to “P value”, as it presents P value for statistical inference instead of the real probability of the events happening.

Fig 4A and Fig 4B seem in the wrong positions.

**Summary and General Comments**

Reviewer #1: I read this paper with great interest. It is well written and produces maps and estimates of spatial autocorrelation for STH and malaria coinfections on the African continent. More detail is needed on the methods. The purpose of the analysis must be made clearer, as the environmental factors that might lead to overlap in unclear (given that the sets of infections have vastly different transmission routes, and this was not discussed). The limitations of the analysis need to be better described.

ABSTRACT

Lines 42-43: P. falciparum is repeated.

The abstract needs to directly state the lowest resolution (km) where coinfection estimation was possible. Also, clarify in the abstract whether STH prevalence was georeferenced when estimated or if the implementation unit for STH prevalence was used and then a point estimate was created in the middle of that implementation unit. And, directly state the age range at which these coinfection estimates are applicable.

INTRO

Throughout the introduction, provide prevalence estimates with their relevant administrative unit.

Given the vastly different transmission routes and the localized influence of soil conditions on STH infections, please expand the introduction to provide a background of exactly what environmental factors, including their directionality of influence, overlap between the two sets of infections. Reporting co-endemicity by country is too vague. Literature on the epidemiology of the two infections is needed and past empirical findings should be noted.

Lines 107-109: Clarify whether these estimates are at the country level.

Reviewer #2: Using data from MAP and ESPEN and methods of spatial cluster analysis, the authors identified high co-infection areas of malaria and each species of STH. The research topic of the manuscript is meaningful for integrated control programs for malaria and STHs. However, there are several important points regarding to the methods and the results needed to address, as described in the above “Methods” and “Results”. Besides, the manuscript seems not well organized, and concerns about the title and the writing should also be paid attention.

Regarding to the title, the “burden” in the title seems not proper as the manuscript mainly provided results for high co-infection areas, instead of the estimation of disease burden due to co-infection. For writing concerns, please refer to the above “Editorial and Data Presentation Modifications”.

PLOS authors have the option to publish the peer review history of their article (what does this mean?). If published, this will include your full peer review and any attached files.

Reviewer #1: No

Reviewer #2: No
---

## [Decision Letter · Decision Letter 1]

23 Aug 2022

Dear Dr. Afolabi,

Thank you very much for submitting your manuscript "Prevalence and distribution pattern of malaria and soil-transmitted helminth co-infections in sub-Saharan Africa, 2000-2018: a geospatial analysis" for consideration at PLOS Neglected Tropical Diseases. As with all papers reviewed by the journal, your manuscript was reviewed by members of the editorial board and by several independent reviewers. The reviewers appreciated the attention to an important topic. Based on the reviews, we are likely to accept this manuscript for publication, providing that you modify the manuscript according to the review recommendations. 

Sincerely,

Uwem Friday Ekpo, PhD

Academic Editor

Marco Coral-Almeida

Section Editor

Reviewer's Responses to Questions

**Key Review Criteria Required for Acceptance?**

**Methods**

-Are the objectives of the study clearly articulated with a clear testable hypothesis stated?

-Is the study design appropriate to address the stated objectives?

-Is the population clearly described and appropriate for the hypothesis being tested?

-Is the sample size sufficient to ensure adequate power to address the hypothesis being tested?

-Were correct statistical analysis used to support conclusions?

-Are there concerns about ethical or regulatory requirements being met?

Reviewer #1: (No Response)

Reviewer #2: (No Response)

**Results**

-Does the analysis presented match the analysis plan?

-Are the results clearly and completely presented?

-Are the figures (Tables, Images) of sufficient quality for clarity?

Reviewer #1: (No Response)

Reviewer #2: (No Response)

**Conclusions**

-Are the conclusions supported by the data presented?

-Are the limitations of analysis clearly described?

-Do the authors discuss how these data can be helpful to advance our understanding of the topic under study?

-Is public health relevance addressed?

Reviewer #1: (No Response)

Reviewer #2: (No Response)

**Editorial and Data Presentation Modifications?**

Reviewer #1: (No Response)

Reviewer #2: (No Response)

**Summary and General Comments**

Reviewer #1: The authors have addressed my requests and greatly improved the manuscript; minor changes are needed for the introduction and terminology before acceptance.

Given that the authors are not looking at the same set of people for malaria and STH nor are they looking at the same age range, using the term co-infection may be misleading. Instead, the manuscript should be revised to clearly note geographical co-endemicity. Co-distributions might also be appropriate as already used in the lay summary. Choose one terminology and use throughout the manuscript.

The additional description of shared environmental or other drivers between malaria and STHs is poorly written without clear examples. Clearly state exactly what climate-related factors predicted both malaria and STHs. Also, note what ‘immune modulation’ is responsible for mixed infections; there is not an interaction between malaria and STH that affects susceptibility to infections, so this sentence is misleading. Clarify what socioeconomic factors affect both diseases. There is still no discussion here of the vastly different transmission routes or differences in climatic factors.

Reviewer #2: The authors revised the manuscript according to editorial and reviewers’ comments. It seems much better compared to the original one. Only one comment: The authors used Sartorius et al’s estimates for STH infection instead of the original kriging method. Are these estimates also based on ESPEN portal? Please clarify it.

PLOS authors have the option to publish the peer review history of their article (what does this mean?). If published, this will include your full peer review and any attached files.

Reviewer #1: No

Reviewer #2: No

Figure Files:

Data Requirements:

Reproducibility:

References

---

## [Decision Letter · Decision Letter 2]

16 Sep 2022

Dear Dr. Afolabi,

We are pleased to inform you that your manuscript 'Prevalence and distribution pattern of malaria and soil-transmitted helminth co-endemicity in sub-Saharan Africa, 2000-2018: a geospatial analysis' has been provisionally accepted for publication in PLOS Neglected Tropical Diseases.

Best regards,

Uwem Friday Ekpo, PhD

Academic Editor

Marco Coral-Almeida

Section Editor

Reviewer's Responses to Questions

**Key Review Criteria Required for Acceptance?**

**Methods**

-Are the objectives of the study clearly articulated with a clear testable hypothesis stated?

-Is the study design appropriate to address the stated objectives?

-Is the population clearly described and appropriate for the hypothesis being tested?

-Is the sample size sufficient to ensure adequate power to address the hypothesis being tested?

-Were correct statistical analysis used to support conclusions?

-Are there concerns about ethical or regulatory requirements being met?

Reviewer #1: (No Response)

**Results**

-Does the analysis presented match the analysis plan?

-Are the results clearly and completely presented?

-Are the figures (Tables, Images) of sufficient quality for clarity?

Reviewer #1: (No Response)

**Conclusions**

-Are the conclusions supported by the data presented?

-Are the limitations of analysis clearly described?

-Do the authors discuss how these data can be helpful to advance our understanding of the topic under study?

-Is public health relevance addressed?

Reviewer #1: (No Response)

**Editorial and Data Presentation Modifications?**

Reviewer #1: (No Response)

**Summary and General Comments**

Reviewer #1: (No Response)

PLOS authors have the option to publish the peer review history of their article (what does this mean?). If published, this will include your full peer review and any attached files.

Reviewer #1: No

---

## [Editor Report · Acceptance letter]

21 Sep 2022

Dear Dr. Afolabi,

We are delighted to inform you that your manuscript, "Prevalence and distribution pattern of malaria and soil-transmitted helminth co-endemicity in sub-Saharan Africa, 2000-2018: a geospatial analysis," has been formally accepted for publication in PLOS Neglected Tropical Diseases.

Best regards,

Shaden Kamhawi

co-Editor-in-Chief

Paul Brindley

co-Editor-in-Chief
